# Open-Vocabulary Object Detection via Language Hierarchy

**Jiaxing Huang, Jingyi Zhang, Kai Jiang, Shijian Lu**[*]
College of Computing and Data Science
Nanyang Technological University, Singapore

## Abstract

Recent studies on generalizable object detection have attracted increasing attention with additional weak supervision from large-scale datasets with image-level labels. However, weakly-supervised detection learning often suffers from image-to-box label mismatch, i.e., image-level labels do not convey precise object information. We design Language Hierarchical Self-training (LHST) that introduces language hierarchy into weakly-supervised detector training for learning more generalizable detectors. LHST expands the image-level labels with language hierarchy and enables co-regularization between the expanded labels and self-training. Specifically, the expanded labels regularize self-training by providing richer supervision and mitigating the image-to-box label mismatch, while self-training allows assessing and selecting the expanded labels according to the predicted reliability. In addition, we design language hierarchical prompt generation that introduces language hierarchy into prompt generation which helps bridge the vocabulary gaps between training and testing. Extensive experiments show that the proposed techniques achieve superior generalization performance consistently across 14 widely studied object detection datasets.

## 1 Introduction

Object detection aims to locate and identify objects in images by providing basic visual information of "where and what objects are". Thanks to the recent advances of deep neural networks, it has achieved great success with various applications in autonomous driving [1, 2, 3, 4], intelligent surveillance [5, 6, 7, 8], wildlife tracking [9, 10, 11], etc. However, learning a generalizable object detector for various downstream tasks that have different data distributions and data vocabularies remains an open research challenge. To this end, weakly-supervised object detection (WSOD) [12, 13, 14, 15], which allows access of large-scale image-level datasets (e.g., ImageNet-21K [16] with 14M images of 21K classes) with super rich data distributions and data vocabularies, has reignited new research interest under the context of learning generalizable detectors.

While exploiting WSOD to learn generalizable detectors, one typical challenge is that the provided image-level labels do not convey precise object information [15] and often mismatch with box-level labels. Recent methods address this challenge by designing various label-to-box assignment strategies that assign the image-level labels to the predicted top-score [13, 14] or max-size [15] object proposals. However, the mismatch problem remains due to the restriction of the raw image-level labels [17]. At the other end, self-training [18, 19, 20] with the detectors pre-trained with [13, 14, 15] can generate box-level pseudo labels without the restriction of image-level labels. It allows learning from more object proposals without the image-to-box label mismatch issue, but it does not benefit much from the provided image-level label supervision.

---

[*]Corresponding author

38th Conference on Neural Information Processing Systems (NeurIPS 2024).

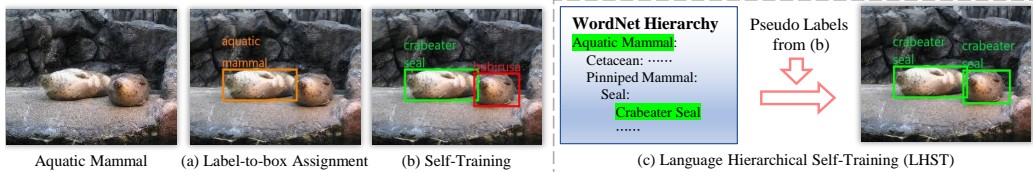

| Aquatic Mammal | (a) Label-to-box Assignment | (b) Self-Training | (c) Language Hierarchical Self-Training (LHST) |

Figure 1: Image-level labels in large-scale datasets such as ImageNet-21k [16] often do not convey precise object information [17, 15] which affects while learning generalizable detectors. Recent methods tackle this issue by various label-to-box assignment strategies [12, 13, 14, 15] as in (a) but are heavily restricted by raw image-level labels and still suffer from image-to-box label mismatch [17]. Self-training [18] with the detectors pre-trained with [13, 14, 15] could circumvent the label mismatch issue but the generated pseudo box labels are error-prone due to the lack of proper supervision as in (b). Our proposed LHST introduces language hierarchy to expand the image-level labels and enables co-regularization between the expanded labels and self-training which allows producing more accurate pseudo box labels in (c).

We propose to incorporate image-level supervision with self-training for learning generalizable detectors, aiming to benefit from self-training while effectively making use of image-level weak supervision. We start from a simple observation: the image-to-box label mismatch largely comes from the ambiguity in language hierarchy, e.g., the image-level label *Aquatic Mammal* in Figure 1 can cover different object-level labels such as seals, dolphins, walruses, etc. With the above observations, we design a **Det**ector with **L**anguage **H**ierarchy (DetLH) that combines language hierarchical self-training (LHST) and language hierarchical prompt generation (LHPG) for learning generalizable detectors.

LHST introduces WordNet's language hierarchy [21] to expand the image-level labels and accordingly enables co-regularization between the expanded labels and self-training. Specifically, the expanded labels are not all reliable though they can mitigate the image-to-box label mismatch problem by providing richer supervision. Here self-training can predict reliability scores for the expanded labels for better selection or weightage of the expanded labels. At the other end, self-training with pseudo box labels allows learning from more proposals and can circumvent the image-to-box label mismatch , but the box-level pseudo labels are usually noisy and may lead to learning degradation [15]. Here the expanded labels provide richer and more flexible supervision which can effectively help suppress prediction noises in self-training.

LHPG helps bridge the vocabulary gaps between training and testing by introducing WordNet's language hierarchy into prompt generation process. Specifically, LHPG leverages the CLIP language encoder [22] to measure the embedding distances between test concepts and WordNet synsets, and then generates the prompt for a given test concept from its best matched WordNet synset. In this way, the test prompts generated by LHPG have been standardized by WordNet and are well aligned with our proposed detector that is trained with WordNet information via LHST. In another word, the combination of LHST and LHPG actually leverages WordNet as a standard and intermediate vocabulary that bridges the gaps between training and testing vocabularies, generating better prompts and leading to better detection performance on downstream applications.

The main contributions of this work are threefold. *First*, we propose language hierarchical self-training that incorporates language hierarchy with self-training for weakly-supervised object detection. *Second*, we design language hierarchical prompt generation, which introduces language hierarchy into prompt generation to bridge the vocabulary gaps between detector training and testing. *Third*, extensive experiments show that our DetLH achieves superior generalization performance consistently across 14 detection benchmarks.

## 2   Related Work

**Weakly-supervised object detection (WSOD)** aims to train object detectors using image-level supervision. Traditional WSOD methods [23, 24, 25, 26, 27] use image-level annotations only without any box annotations and thus focus on low-level proposal mining techniques [28, 29, 12, 30, 31, 32], leading to unsatisfying localization performance. **Semi-supervised WSOD** [33, 34, 35, 36, 37, 38, 39]

has been proposed to further improve the performance, which leverages both box-level and image-level annotated data. With better localization quality, recent methods [13, 14, 15, 40] design various label-to-box assignment strategies, such as assigning image-level labels to max-score anchors [13], max-score proposals [14] or max-size proposals [15]. Our work belongs to semi-supervised WSOD. Different from previous methods, we tackle the image-to-box label mismatch by introducing language hierarchy into self-training.

**Large-vocabulary object detection** [41, 13, 42, 43, 44] researches on detecting thousands of categories. Most previous papers focus on tackling the long-tail issue [45, 46, 47, 48, 49, 50], e.g., by using equalization losses [51, 52], SeeSaw loss [53], or Federated Loss [54]. Recent semi-supervised WSOD methods [13, 14, 15] and our work circumvent the long-tail problem by leveraging more balanced image-level datasets such as ImageNet-21K.

**Open-vocabulary object detection** focuses on detecting objects conditioned on arbitrary words (i.e., any category names). A common strategy [55, 56, 57, 58, 59] is to replace the detector's classification layer with the language embeddings of category names. Recent methods [60, 61, 62, 63, 17, 15] leverage the powerful CLIP [22] model by using its text embeddings [60, 61, 62, 63, 17, 15] or conducting knowledge distillation [60, 63, 17]. Similar to Detic [15], our work uses CLIP text embeddings as the classifier and leverages image-level annotated data instead of distilling knowledge from CLIP.

**Language hierarchy** has been widely studied for visual recognition tasks [64], especially for large-vocabulary visual recognition. Most existing studies [65, 66, 67] focus on image classification tasks, e.g., leveraging language hierarchy for multi-label image classification [65, 66, 67, 68, 69, 70, 71], modelling hierarchical relations among classes [68, 69] or facilitating classification training [70, 71]. Different from previous work, we introduce language hierarchy into self-training for weakly-supervised object detection.

# 3 Method

This work focuses on learning generalizable object detectors via weakly-supervised detector training [15], which leverages additional large-scale image-level datasets to enlarge the data distributions and data vocabularies in detector training. We first describe the task definition with training and evaluation setups. Then, we present our proposed DetLH which is detailed in two major aspects on Language Hierarchical Self-training (LHST) that introduces language hierarchy into detector training, and Language Hierarchical Prompt Generation (LHPG) that introduces language hierarchy into prompt generation.

## 3.1 Task Definition

**Training setup.** The training data consists of two parts: 1) a detection dataset $\mathcal{D}_{det} = \{(x, y_{det})_i\}_{i=1}^{|D_{det}|}$, where $x$ denotes an image while $y_{det}$ stands for the class and bounding box labels for $x$; 2) an image classification dataset $\mathcal{D}_{cls} = \{(x, y_{cls})_i\}_{i=1}^{|D_{cls}|}$ where $y_{cls}$ denotes the image-level label (i.e., a one-hot vector) for $x$. Given the two datasets, the goal is to learn a generalizable detection model $F$ by jointly optimizing $F$ over $\mathcal{D}_{det}$ and $\mathcal{D}_{cls}$:

$$Loss = \sum_{(x, y_{det}) \in \mathcal{D}_{det}} \mathcal{L}_{det}(F(x), y_{det}) + \sum_{(x, y_{cls}) \in \mathcal{D}_{cls}} \mathcal{L}_{weak}(F(x), y_{cls}), \qquad (1)$$

where $\mathcal{L}_{det}(\cdot) = \mathcal{L}_{rpn}(\cdot) + \mathcal{L}_{reg}(\cdot) + \mathcal{L}_{cls}(\cdot)$ is the fully-supervised detection loss function while $\mathcal{L}_{rpn}(\cdot)$, $\mathcal{L}_{reg}(\cdot)$, and $\mathcal{L}_{cls}(\cdot)$ denote RPN, Regression, and Classification loss functions, respectively. $\mathcal{L}_{weak}$ is the weakly-supervised loss function to train detectors with image-level labels.

**Evaluation setup.** As the goal is to learn a generalizable detection model that works well on various unseen downstream tasks, we conduct zero-shot cross-dataset evaluation[2] to assess the generalization performance of the trained detection model. Note, different domain adaptation [72, 73, 74, 75] that generally uses downstream data in training, our setup is similar to domain generalization [76, 77] that does not involve downstream data in training.

---

[2]zero-shot cross-dataset evaluation here means that the model is evaluated on unseen datasets, which is the same as the one defined in CLIP [22].

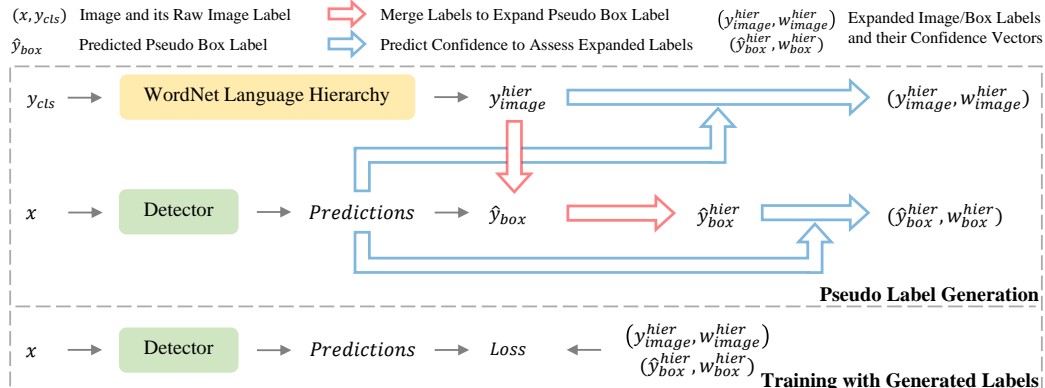

Figure 2: The proposed language hierarchical self-training consists of two flows including Pseudo Label Generation (top box) and Training with Generated Labels (bottom box). The Pseudo Label Generation flow leverages WordNet to expand the image-level labels, and then merges the expanded image-level labels with the predicted pseudo box labels, such that the expanded image-level labels could provide richer and more flexible supervision (than the limited and rigid raw labels) to regularize the self-training which is prone to errors in pseudo labeling. In addition, as the labels expanded by WordNet (i.e., the expanded logits '1' in $y_{image}^{hier}$ and $y_{box}^{hier}$) are not all reliable, Pseudo Label Generation predicts reliability scores for the expanded labels to adaptively re-weight them when applying them on different images or pseudo boxes. In Training with Generated Labels, we optimize the detector with the generated image-level and box-level labels, where the image-level training could regularize the training with pseudo box-level labels as pseudo box labels vary along training iterations and are not very stable.

**Open-vocabulary Detector.** We modify the classification layer of the detector into an open-vocabulary format such that the detector could be tested over unseen datasets. Specifically, we replace the weights of the detector's classification layer with the fixed language embeddings encoded from class names, where the object classification could be achieved by matching the object's embedding and the fixed language embeddings. We adopt the CLIP language embeddings [22] as the classification weights as in [15, 60]. In this way, the modified detector could theoretically detect any target concepts on any target data. As reported in [15], detectors trained solely on detection datasets often exhibits constrained performance due to the small-scale training images and vocabularies. Similar to [15], our proposed DetLH introduces large-scale image-level datasets to enlarge the data distributions and data vocabularies in detector training, leading to more generalizable detectors and better generalization performance on various unseen datasets.

### 3.2 Language Hierarchical Self-training

The proposed LHST utilizes WordNet's language hierarchy to expand the image-level labels, which enables co-regularization between the expanded image-level labels and self-training as illustrated in Figure 2.

**Overview.** For *fully supervised detector training* over the detection dataset, we feed box-level annotated samples $(x, y_{det}) \in \mathcal{D}_{det}$ to the detection model $F$ and optimize $F$ with the standard fully supervised detection loss, i.e., the first term of Eq. 1. For *weakly-supervised detector training* over the image-level annotated dataset $(x, y_{cls}) \in \mathcal{D}_{cls}$ shown in Figure 2, we first leverage WordNet's language hierarchy to expand the raw image-level label $y_{cls}$ into $y_{image}^{hier}$ (the hierarchical image-level label), and merge $y_{image}^{hier}$ and the generated pseudo box label $\hat{y}_{box}$ to acquire $\hat{y}_{box}^{hier}$ (the hierarchical box-level pseudo label). Then, we optimize the detector with $(\hat{y}_{box}^{hier}, w_{box}^{hier})$ and $(y_{image}^{hier}, w_{image}^{hier})$, where $w_{image}^{hier}$ and $w_{box}^{hier}$ denote the predicted reliability scores of the expanded logits '1' in $y_{image}^{hier}$ and $y_{box}^{hier}$ and are used to weight the labels in loss calculation.

**Expanding image labels with language hierarchy.** Given image-level annotated dataset $(x, y_{cls}) \in \mathcal{D}_{cls}$ ($y_{cls}$ is a label vector with length $C$ and $C$ denotes the number of classes), we leverage WordNet's class name hierarchy [21] to expand $y_{cls}$ into $y_{image}^{hier}$ as the following:

$$y_{image}^{hier} = \text{WordNet}(y_{cls}), \tag{2}$$

where the function $\text{WordNet}(\cdot)$ recursively finds all hypernyms and hyponyms of the input (i.e., the class indicated in $y_{cls}$) and sets their positions in the label vector $y_{cls}$ to be '1' to expand $y_{cls}$ into $y_{image}^{hier}$. In this way, a single-label annotation could be expanded into a multi-label annotation within the very rich ImageNet-21K vocabulary.

**Generating pseudo box labels with predictions.** Given the image $x \in \mathcal{D}_{cls}$, we feed $x$ into the detector $F$ to acquire the prediction as following:

$$\{p_n^c\}_{1 \le n \le N, 1 \le c \le C} = F(x), \tag{3}$$

where $p_n$ is the probability vector of the predicted $n$-th bounding box after Softmax, and $p_n^c$ denotes the predicted $c$-th category probability. Note we filter out a prediction if its max confidence score is lower than the threshold $t$, and $N$ denotes the number of predicted object proposals after filtering, i.e., $max(\{p_n^c\}_{1 \le c \le C}) \ge t, \forall n$.

Then the pseudo category label $\hat{y}_{box} = \{\hat{y}_n\}_{1 \le n \le N}$ for $N$ boxes in image $x$ is derived by:

$$\underset{\hat{y}_n}{\arg\max} \sum_{c=1}^{C} \hat{y}_n^c \log p_n^c, \ s.t. \ \hat{y}_n \in \Delta^C, \forall n, \tag{4}$$

where $\hat{y}_n = (\hat{y}_n^{(1)}, \hat{y}_n^{(2)}, ..., \hat{y}_n^{(C)})$ is the predicted category label, and $\Delta^C$ denotes a probability simplex with length $C$.

**Merging image and pseudo box labels.** As the predicted pseudo box label $\hat{y}_{box}$ is error-prone, we regularize it with the expanded image-level supervision by merging $y_{image}^{hier}$ and $\hat{y}_{box}$ as the following:

$$\hat{y}_{box}^{hier}(n) = \hat{y}_{box}(n) \lor y_{image}^{hier}, \forall n, \tag{5}$$

where $\lor$ denotes the logical "OR" operator.

**Assessing the expanded labels.** As the labels expanded by WordNet (i.e., the expanded logits '1' in $\hat{y}_{box}^{hier} = \{\hat{y}_n^c\}_{1 \le n \le N, 1 \le c \le C}$) are not all reliable, we predict a reliability score $w_n^c$ for the expanded label to adaptively re-weight $y_n^c \in \hat{y}_{box}^{hier}$ when applying it on different pseudo boxes. We measure the reliability of $y_n^c$ with prediction $p_n^c$, and $w_{box}^{hier} = \{w_n^c\}_{1 \le n \le N, 1 \le c \le C}$ can be derived by:

$$w_n^c = \begin{cases} p_n^c & \text{if } y_{image}^{hier}{}^{(c)} \neq y_{cls}^{(c)} \\ 1 & \text{otherwise,} \end{cases} \tag{6}$$

where $y_{image}^{hier}{}^{(c)} \neq y_{cls}^{(c)}$ returns True if the $c$-th label logit in $y_{image}^{hier}$ is expanded by WordNet, which also applies to $\hat{y}_{box}^{hier}$ as $\hat{y}_{box}^{hier}$ is expanded by mergeing it with $y_{image}^{hier}$.

Given the prediction $\{p_n^c\}_{1 \le n \le N, 1 \le c \le C}$, the merged pseudo box label $\hat{y}_{box}^{hier} = \{\hat{y}_n^c\}_{1 \le n \le N, 1 \le c \le C}$ and its reliability score $w_{box}^{hier} = \{w_n^c\}_{1 \le n \le N, 1 \le c \le C}$, we optimize the detector $F$ as the following:

$$\mathcal{L}_{box}(F(x)) = \sum_{n}^{N} \sum_{c}^{C} (\text{BCE}(p_n^c, y_n^c) \times w_n^c), \tag{7}$$

where $\text{BCE}(\cdot)$ denotes the binary cross-entropy loss.

In addition, training with the predicted pseudo box labels is not very stable as pseudo box labels vary along training process. Thus, we regularize the training of $\mathcal{L}_{box}(F(x))$ with an image-level loss defined as the following:

$$\mathcal{L}_{image}(F(x)) = \sum_{c}^{C} (\text{BCE}(p_{image}^c, y_{image}^{hier}{}^{(c)}) \times w_{image}^c), \tag{8}$$

where $p_{image} = \{p_{image}^c\}_{1 \le c \le C}$ denotes the category probability predicted for the image-level proposal. $w_{image}^{hier} = \{w_{image}^c\}_{1 \le c \le C}$ denotes the reliability score for the expanded logits "1" in

$y_{image}^{hier}$. Similar to Eq. 6, $w_{image}^c = p_{image}^c$ if $y_{image}^{hier\ (c)} \neq y_{cls}^{(c)}$, otherwise $w_{image}^c = 1$. Besides, $\text{BCE}(\cdot)$ denotes the binary cross-entropy loss.

**Training objective.** The overall training objective of Language Hierarchical Self-training is defined as:

$$\mathcal{L}_{lhst} = \sum_{(x, y_{det}) \in \mathcal{D}_{det}} \mathcal{L}_{det}(F(x), y_{det}) + \sum_{(x, y_{cls}) \in \mathcal{D}_{cls}} \left( \mathcal{L}_{box}(F(x)) + \mathcal{L}_{image}(F(x)) \right) \quad (9)$$

**Language Hierarchical Prompt Generation.** As the goal is to learn a generalizable detection model that works well on various downstream tasks, one typical challenge is the vocabulary gap between detector training datasets (i.e., LVIS and ImageNet-21K) and detector testing datasets (e.g., object365 or customized data). A common solution of tackling the vocabulary gaps is to conduct prompt learning [78] to generate proper category prompts. However, prompt learning generally requires labeled target images for additional training.

In this work, we tackle the vocabulary gaps by generating prompts with the help of WordNet, which introduces little computation overhead and does not require labeled target images and additional training. To this end, we design language hierarchical prompt generation (LHPG) that works by incorporating WordNet information into prompt generation process. Specifically, LHPG leverages CLIP language encoder [22] to measure the embedding distances between test concepts and WordNet synsets, and then generates the prompt for a given test concept from its best matched WordNet synset: $V_{test}^{\text{WordNet}} = \text{CLIP}(V_{test}, \text{WordNet})$, where $V_{test}$ denotes test vocabulary, WordNet denotes WordNet synsets, CLIP denotes CLIP language encoder and $V_{test}^{\text{WordNet}}$ stands for the best matched WordNet synsets for the classes in $V_{test}$. Then, we generate test prompts from $V_{test}^{\text{WordNet}}$. As compared with $V_{test}$, our $V_{test}^{\text{WordNet}}$ has been standardized by WordNet and is well aligned with our proposed detector that is trained with WordNet information via LHST. In another word, the combination of LHST and LHPG makes use of WordNet as a standard and intermediate vocabulary that bridges the gaps between training and testing vocabularies, generating better prompts and leading to better detection performance on downstream applications.

## 4 Experiments

We evaluate our DetLH on 14 widely adopted detection benchmarks. We follow the zero-shot cross-dataset object detection setting proposed in [17, 15]. More details like **Dataset** and **Implementation Details** are provided in the appendix.

Table 1: **Zero-shot cross-dataset object detection for common objects.** All detectors are trained over the training datasets (LVIS and ImageNet-21K) and evaluated over target datasets (i.e., Object365 and Pascal VOC with objects from common classes and scenarios) without finetuning. "Dataset-specific oracles" denote the detectors that are fully supervised which are trained by using the training data of respective datasets.

| Method | Object365 [79] | | | | | | Pascal VOC [80] | | | | | |
|---|---|---|---|---|---|---|---|---|---|---|---|---|
| | AP | AP50 | AP75 | APs | APm | APl | AP | AP50 | AP75 | APs | APm | APl |
| WSDDN [12] | 21.0 | 29.1 | 22.7 | 8.7 | 20.9 | 31.2 | 61.6 | 82.7 | 67.5 | 24.8 | 50.9 | 73.5 |
| YOLO9000 [13] | 21.0 | 28.5 | 22.6 | 8.6 | 20.7 | 30.9 | 62.6 | 83.6 | 68.7 | 23.7 | 52.0 | 73.9 |
| DLWL [14] | 21.3 | 29.1 | 23.0 | 8.8 | 21.0 | 31.5 | 62.4 | 83.4 | 68.3 | 23.8 | 51.2 | 73.8 |
| Detic [15] | 21.6 | 29.4 | 23.4 | 9.0 | 21.4 | 31.9 | 62.4 | 83.3 | 68.5 | 23.7 | 51.8 | 73.9 |
| **DetLH (Ours)** | **23.6** | **32.5** | **25.5** | **9.8** | **23.5** | **35.0** | **64.4** | **86.1** | **70.8** | **25.3** | **54.1** | **75.3** |
| Dataset-specific oracles | 31.2 | - | - | - | - | - | 54.4 | 79.7 | 59.1 | 19.0 | 40.8 | 64.5 |

### 4.1 Comparison with the state-of-the-art

We conduct extensive experiments to benchmark our proposed DetLH with state-of-the-art methods. We evaluate them on 14 widely studied object detection datasets to assess their zero-shot cross-dataset generalization ability. Tables 1- 5 report zero-shot cross-dataset detection results for common objects, autonomous driving, intelligent surveillance, and wildlife detection, respectively. More details are to be described in the following paragraphs.

Table 2: **Zero-shot cross-dataset object detection for autonomous driving.** All detectors are trained over the training datasets (LVIS and ImageNet-21K) and evaluated over autonomous driving datasets (i.e., Cityscapes, Vistas and SODA10M) without finetuning.

| Method | Cityscapes [1] | | | Vistas [2] | | | SODA10M [3] | | | Average | | |
|---|---|---|---|---|---|---|---|---|---|---|---|---|
| | AP | AP50 | AP75 | AP | AP50 | AP75 | AP | AP50 | AP75 | AP | AP50 | AP75 |
| WSDDN [12] | 28.2 | 45.4 | 27.1 | 22.3 | 34.0 | 23.3 | 17.4 | 28.9 | 17.1 | 22.6 | 36.1 | 22.4 |
| YOLO9000 [13] | 28.8 | 46.2 | 27.4 | 22.5 | 34.6 | 23.4 | 18.3 | 30.4 | 18.0 | 23.2 | 37.0 | 22.9 |
| DLWL [14] | 28.6 | 45.6 | 28.1 | 22.5 | 34.7 | 23.2 | 18.3 | 30.4 | 18.0 | 23.1 | 36.9 | 23.0 |
| Detic [15] | 29.6 | 47.1 | 28.4 | 23.0 | 35.6 | 23.6 | 18.8 | 30.9 | 18.5 | 23.8 | 37.9 | 23.5 |
| **DetLH (Ours)** | **31.2** | **50.3** | **29.1** | **26.5** | **44.0** | **25.8** | **25.1** | **38.4** | **26.1** | **27.6** | **44.2** | **27.0** |
| Dataset-specific oracles | 43.0 | 69.0 | 42.6 | 28.1 | 45.8 | 28.5 | 44.7 | 68.2 | 47.3 | 38.6 | 61.0 | 39.5 |

Table 3: **Zero-shot cross-dataset object detection under different weather and time-of-day conditions (using metric AP50).** All detectors are trained over the training datasets (LVIS and ImageNet-21K) and evaluated over BDD100K and DAWN datasets without finetuning.

| Method | BDD100K-weather [81] | | | | | | BDD100K-time-of-day [81] | | | | DAWN [82] | | | Avg |
|---|---|---|---|---|---|---|---|---|---|---|---|---|---|---|
| | rainy | snowy | overcast | cloudy | foggy | undefined | daytime | dawn&dusk | night | undefined | fog | sand | snow | |
| WSDDN [12] | 35.0 | 33.0 | 38.3 | 41.7 | 26.7 | 46.0 | 39.1 | 35.9 | 26.6 | 50.2 | 62.6 | 55.0 | 65.6 | 42.8 |
| YOLO9000 [13] | 34.4 | 33.6 | 39.5 | 41.8 | 31.0 | 45.4 | 39.6 | 35.9 | 28.8 | 46.6 | 60.6 | 53.9 | 64.4 | 42.7 |
| DLWL [14] | 34.8 | 33.4 | 38.8 | 43.8 | **40.2** | 45.2 | 40.1 | 35.1 | 28.7 | 45.0 | 62.1 | 56.1 | 63.7 | 43.6 |
| Detic [15] | 34.3 | 33.2 | 39.5 | 41.9 | 27.9 | 45.4 | 39.2 | 35.5 | 28.8 | 48.2 | 52.3 | 54.1 | 56.1 | 41.3 |
| **DetLH (Ours)** | **40.2** | **37.5** | **48.2** | **49.3** | 37.1 | **49.9** | **45.7** | **40.0** | **34.2** | **53.0** | **63.2** | **57.6** | **67.3** | **47.9** |
| Dataset-specific oracles | 52.0 | 52.5 | 56.3 | 56.3 | 21.3 | 65.4 | 57.0 | 50.4 | 48.6 | 27.7 | 56.7 | 48.4 | 26.4 | 47.6 |

Table 4: **Zero-shot cross-dataset object detection for intelligent surveillance.** All detectors are trained over the training datasets (LVIS and ImageNet-21K) and evaluated over surveillance datasets MIO-TCD, BAAI-VANJEE, DETRAC and UAVDT without finetuning.

| Method | MIO-TCD [5] | | | BAAI-VANJEE [6] | | | DETRAC [7] | | | UAVDT [8] | | | Average | | |
|---|---|---|---|---|---|---|---|---|---|---|---|---|---|---|---|
| | AP | AP50 | AP75 | AP | AP50 | AP75 | AP | AP50 | AP75 | AP | AP50 | AP75 | AP | AP50 | AP75 |
| WSDDN [12] | 11.3 | 17.6 | 11.6 | 13.1 | 19.6 | 13.3 | 25.6 | 35.3 | 31.1 | 17.1 | 31.9 | 16.0 | 16.7 | 26.1 | 18.0 |
| YOLO9000 [13] | 12.7 | 19.7 | 13.0 | 13.1 | 19.3 | 13.0 | 29.1 | 39.4 | 35.3 | 18.6 | 33.9 | 17.7 | 18.3 | 28.1 | 19.7 |
| DLWL [14] | 12.9 | 20.1 | 12.9 | 13.5 | 20.0 | 13.6 | 27.8 | 38.0 | 33.6 | 16.6 | 31.1 | 15.1 | 16.7 | 26.1 | 18.0 |
| Detic [15] | 13.4 | 20.6 | 13.9 | 16.9 | 23.6 | 17.6 | 28.7 | 39.2 | 34.8 | 18.6 | 34.2 | 17.6 | 19.4 | 29.4 | 21.0 |
| **DetLH (Ours)** | **15.8** | **24.5** | **16.0** | **17.9** | **25.1** | **18.5** | **32.7** | **44.0** | **39.7** | **20.1** | **36.6** | **19.3** | **21.6** | **32.6** | **23.4** |
| Dataset-specific oracles | 45.2 | 63.1 | 50.8 | 40.6 | 58.6 | 43.7 | 53.1 | 70.6 | 63.5 | 33.8 | 60.4 | 35.2 | 43.2 | 63.2 | 48.3 |

**Object detection for common objects.** Table 1 shows that DetLH outperforms state-of-the-art methods clearly on common object datasets Object365 and Pascal VOC. In addition, we can observe that DetLH even brings significant gains above the *dataset-specific oracle* (i.e., the model that is fully trained on the target training data) on Pascal VOC (i.e., a small-scale dataset), showing the advantages of leveraging large-scale training data.

**Object detection for autonomous driving.** As shown in Table 2, our DetLH outperforms state-of-the-art methods by large margins on various autonomous driving datasets, showing that DetLH still works effectively while facing large variations in camera views from autonomous driving scenarios to the base-dataset scenarios (LVIS and ImageNet-21K), e.g., autonomous driving images are captured under very different camera views. In addition, the experimental results in Table 3 show that our DetLH brings significant performance gains against state-of-the-art methods when encountering various weather and time-of-day conditions, which demonstrates the effectiveness of DetLH while detecting objects under large noises [83], e.g., the images captured under different weather and time-of-day conditions may have very different styles and image quality.

**Object detection for intelligent surveillance.** From Table 4, we can observe that our DetLH outperforms state-of-the-art methods by clear margins on various intelligent surveillance datasets, indicating that DetLH is also tolerant to large changes in the camera lens and angles which often happen to intelligent-surveillance images that are captured under very different camera lens and angles (e.g., surveillance cameras are often with the wide-angle lens and used in high angle views).

**Object detection for Wildlife.** The experimental results in Table 5 show that our DetLH performs well on various wildlife detection datasets, showing that DetLH works effectively for detecting fine-grained categories that exist widely in wildlife detection datasets. The significant performance

Table 5: **Zero-shot cross-dataset object detection for Wildlife Detection.** All detectors are trained over the training datasets (LVIS and ImageNet-21K) and evaluated over wildlife datasets (i.e., Arthropod Detection, AfricanWildlife and Animals Detection) without finetuning.

| Method | Arthropod Detection [9] | | | AfricanWildlife [10] | | | Animals Detection [11] | | | Average | | |
|---|---|---|---|---|---|---|---|---|---|---|---|---|
| | AP | AP50 | AP75 | AP | AP50 | AP75 | AP | AP50 | AP75 | AP | AP50 | AP75 |
| WSDDN [12] | 18.1 | 26.3 | 18.8 | **76.7** | **88.2** | **84.0** | 36.0 | 41.7 | 37.5 | 43.6 | 52.0 | 46.7 |
| YOLO9000 [13] | 22.6 | 33.2 | 22.5 | 75.9 | 87.5 | 83.2 | 39.0 | 45.4 | 40.8 | 45.8 | 55.3 | 48.8 |
| DLWL [14] | 25.3 | 34.7 | 26.3 | 74.7 | 86.2 | 81.3 | 41.7 | 48.1 | 43.7 | 47.2 | 56.3 | 50.4 |
| Detic [15] | 27.4 | 36.7 | 29.2 | 68.9 | 80.9 | 76.4 | 41.1 | 47.7 | 42.9 | 45.8 | 55.1 | 49.5 |
| **DetLH (Ours)** | **36.2** | **49.0** | **38.3** | 74.8 | 87.2 | 81.8 | **44.3** | **51.2** | **46.3** | **51.8** | **62.5** | **55.5** |
| Dataset-specific oracles | 75.1 | 86.3 | 79.9 | 82.7 | 90.9 | 89.1 | 64.4 | 74.6 | 69.4 | 74.1 | 83.9 | 79.5 |

gains largely come from the introduction of language hierarchy into detector training and prompt generation, which helps model the hierarchical relations among parent and fine-grained subcategories and thus leads to better fine-grained object detection.

The superior detection performance of our DetLH is largely attributed to our two core designs, i.e., LHST and LHPG. LHST enables effective usage of large-scale image-level annotated images and significantly enlarges the data distribution and the data vocabulary in detector training, yielding robust performance under large cross-dataset gaps in data distribution and vocabulary. LHPG ingeniously helps mitigate the vocabulary gaps between detector training and testing. It improves the overall confidence of detection and benefits the detection as large data distribution gaps (or large data vocabulary gaps) often lead to low-confidence predictions and poor detection results.

## 4.2 Ablation Studies

We perform ablation studies with Swin-B [84] based CenterNet2 [54] over the large-scale Object365 dataset as shown in Table 6. As the core of our proposed DetLH, we examine how our designed LHST and LHPG contribute to the overall performance of zero-shot cross-dataset object detection. As shown in Table 6, the *baseline* (Box-Supervised [15]) does not perform well as it uses box-level training data only. It can be observed that LHST outperforms the baseline clearly, showing that LHST can effectively leverage the large-scale image-level annotated dataset to significantly enlarge the data distribution and data vocabulary involved in detector training, leading to

Table 6: **Ablation studies of our DetLH** with Language Hierarchical Self-training (LHST) and Language Hierarchical Prompt Generation (LHPG). The experiments are conducted with Swin-B based CenterNet2 [15] and the detectors are evaluated on Object365 in zero-shot cross-dataset object detection setup.

| Method | LHST | LHPG | AP50 |
|---|---|---|---|
| Box-Supervised [15] | | | 26.5 |
| | ✓ | | 31.3 |
| | | ✓ | 31.0 |
| **DetLH (Ours)** | ✓ | ✓ | **32.5** |

much better zero-shot cross-dataset detection performance. In addition, LHPG brings clear performance improvements in zero-shot cross-dataset detection by introducing language hierarchy into prompt generation, demonstrating the effectiveness of LHPG in mitigating the vocabulary gaps between training and testing. Moreover, the inclusion of both LHST and LHPG in the proposed DetLH performs clearly the best, indicating the complementary property of our two designs.

## 4.3 Discussion

Table 7: **Zero-shot cross-dataset object detection on various datasets.** Results are averaged on 14 widely studied datasets.

| Method | Averaged over 14 detection datasets | | | | | |
|---|---|---|---|---|---|---|
| | AP | AP50 | AP75 | APs | APm | APl |
| WSDDN [12] | 29.9 | 42.5 | 31.4 | 14.8 | 25.9 | 44.2 |
| YOLO9000 [13] | 30.9 | 43.8 | 32.4 | 14.1 | 25.8 | 45.1 |
| DLWL [14] | 31.0 | 44.0 | 32.5 | 15.4 | 26.3 | 45.3 |
| Detic [15] | 31.0 | 44.0 | 32.8 | 14.6 | 27.5 | 45.5 |
| **DetLH (Ours)** | **34.6** | **49.3** | **36.4** | **16.0** | **28.4** | **49.5** |

**Generalization across various detection tasks:** We study the generalization of our DetLH by conducting zero-shot cross-dataset object detection on 14 widely studied object detection datasets. Tables 1- 5 show that DetLH achieves superior performance consistently across all the detection applications. Besides, Table 7 summarizes the detection results averaged on 14 datasets, showing that DetLH clearly outperforms the state-of-the-art methods.

**Generalization across various network architectures:** We study the generalization of the proposed DetLH from the perspective of network architectures. Specifically, we perform extensive evaluations with four representative network architectures, including one Transformer-based (i.e., Swin-B) and three CNN-based (i.e., ConvNeXt-T, ResNet-50 and ResNet-18). Experimental results in Table 8 show that the proposed DetLH outperforms the state-of-the-art method consistently over different network architectures.

Table 8: **Zero-shot cross-dataset object detection with different network architectures.** All networks architectures are trained over the training datasets (LVIS and ImageNet-21K) and evaluated over Object365 without finetuning.

| Method | Architecture | Object365 | | | | | |
|---|---|---|---|---|---|---|---|
| | | AP | AP50 | AP75 | APs | APm | APl |
| Detic [15] | Swin-B [84] | 21.6 | 29.4 | 23.4 | 9.0 | 21.4 | 31.9 |
| **DetLH (Ours)** | | 23.6 | 32.5 | 25.5 | 9.8 | 23.5 | 35.0 |
| Detic [15] | ConvNeXt-T [85] | 16.9 | 23.5 | 18.1 | 6.8 | 16.6 | 24.9 |
| **DetLH (Ours)** | | 18.9 | 26.8 | 20.2 | 7.6 | 18.8 | 28.2 |
| Detic [15] | ResNet-50 [86] | 16.2 | 22.8 | 17.5 | 6.3 | 16.2 | 24.1 |
| **DetLH (Ours)** | | 17.7 | 25.5 | 19.0 | 6.9 | 17.9 | 26.4 |
| Detic [15] | ResNet-18 [86] | 10.8 | 15.5 | 11.6 | 3.9 | 10.2 | 16.2 |
| **DetLH (Ours)** | | 11.8 | 17.3 | 12.5 | 4.3 | 11.4 | 17.7 |

**Parameter Studies for Language Hierarchical Self- training (LHST).** In generating pseudo box labels in LHST, we filter out a prediction if its max confidence score is lower than the threshold $t$. We study the threshold $t$ by changing it from $0.65$ to $0.85$ with a step of $0.05$. Table 12 reports the experimental results on zero-shot transfer object detection over object365 dataset. We can observe that the detection performance is not sensitive to the threshold $t$.

Table 9: **Parameter Studies for Language Hierarchical Self- training (LHST)** on zero-shot transfer object detection over object365 dataset. We study the thresholding parameter $t$ used in generating pseudo box labels in LHST.

| Threshold $t$ | 0.65 | 0.70 | 0.75 | 0.80 | 0.85 |
|---|---|---|---|---|---|
| AP50 | 31.1 | 31.3 | 31.3 | 31.3 | 31.2 |

Due to the space limit, we provide more DetLH discussions and visualizations in the appendix.

# 5 Conclusion

This paper presents DetLH, a Detector with Language Hierarchy that combines language hierarchical self-training (LHST) and language hierarchical prompt generation (LHPG) for learning generalizable object detectors. LHST introduces WordNet's language hierarchy to expand the image-level labels and accordingly enables co-regularization between the expanded labels and self-training. LHPG helps mitigate the vocabulary gaps between training and testing by introducing WordNet's language hierarchy into prompt generation. Extensive experiments over multiple object detection tasks show that our DetLH achieves superior performance as compared with state-of-the-art methods. In addition, we demonstrate that DetLH works well with different network architectures such as Swin-B, ConvNeXt-T, ResNet-50, etc. Moving forward, we will explore language hierarchy to further expand the labels in an open-vocabulary manner in addition to the closed ImageNet-21K's vocabulary.

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

# Appendix

## A Dataset and Implementation Details

### A.1 Implementation Details

As in [17, 15], we adopt the CenterNet2 [54] with Swin-B [84] backbone in all the experiments (except for Table 8 where different backbone architectures were used, e.g., ConvNeXt-T, ResNet-50 and ResNet-18). We employ SGD [87] as the optimizer and adopt the cosine learning rate scheduler with a warm-up of 1000 iterations [15]. We set the input sizes of box-level annotated images (i.e., LVIS) and image-level annotated images (i.e., ImageNet-21K) as $896 \times 896$ and $448 \times 448$, respectively. As mentioned in Section 3.1, we employ the CLIP text embeddings [22] as the classifier instead of using the original one in [54]. During training, we sample box-level and image-level mini-batches in a $1:16$ ratio. We set the confidence threshold $t$ (in pseudo box label generation in Eq. 3) as 0.75 in all experiments except in parameter analysis. Note we pre-train the detector over the training datasets (i.e., training on LVIS with the conventional detection loss and on ImageNet-21K with the conventional image classification loss) such that it can generate pseudo box-level labels of 21K classes for self-training.

As described in the main text, we train our detector over two training datasets LVIS and ImageNet-21K, and evaluate the trained detector over 14 evaluation datasets as listed without fine-tuning.

### A.2 Training Dataset

**LVIS** [41] is a large vocabulary dataset designed for long-tailed instance segmentation, which contains 100K images and 1203 categories. LVIS provides high-quality instance-wise annotations, including instance masks, class labels and bounding boxes.

**ImageNet-21K** [16] is a large and diverse dataset over 14M images across more than 21K categories. All categories in ImageNet-21K are defined by WordNet Synsets with clear and accurate definitions and certain language hierarchy.

### A.3 Evaluation Dataset

**Object365** [79] is a large-scale object detection dataset designed for object detection in the wild. This dataset contains 638K images across 365 categories, including 600K images for training and 38K images for validation.

**Pascal VOC** [80] is a real-world dataset with two sub-datasets, *i.e.*, PASCAL VOC 2007 and PASCAL VOC 2012. PASCAL VOC 2007 contains 2,501 training images and 2,510 validation images, and PASCAL VOC 2012 contains 5,717 training images and 5,823 validation images. This dataset provides bounding box annotations with 20 categories.

**Cityscapes** [1] is a dataset designed for the understanding of street scenes. The images in Cityscapes are captured under normal weather conditions from 50 cities, including 2,975 training images and 500 validation images with pixel-wise instance annotations of 8 categories.

**Vista** [2] is a street-level autonomous driving dataset. This dataset contains high-resolution images that cover diverse urban scenes from around the world, including 18K training images and 2K validation images with pixel-wise instance annotations.

**SODA10M** [3] is a large-scale object detection dataset for autonomous driving, which contains 10M unlabeled images and 20K images with bounding box annotations of 6 object categories. The images in this dataset are collected within 27833 driving hours covering a variety time periods and locations across 32 different cities.

**BDD100k** [81] is a large-scale driving video dataset that contains diverse driving scenarios, including different weather conditions (*i.e.*, clear, cloudy, overcast, rainy, snowy and foggy) and times of day (*i.e.*, dawn, daytime and night). This dataset contains 100K videos, including 70K training videos and 10K validation videos with bounding box annotations of 10 categories.

**Arthropod Detection** [9] is a detection dataset for arthropods taxonomy orders identification. The images are collected from a variety of agricultural settings (*e.g.*, fields, greenhouses, warehouses), including over 12K images with bounding box annotations of 7 categories.

**AfricanWildlife** [10] is a detection dataset which contains images of African wildlife with bounding box annotations. This dataset contains 4 different categories of African wildlife including buffalo, elephant, rhino, zebra and each category contains 376 images.

**Animals Detection** [11] is a public dataset of various animals. This dataset contains animal images with bounding boxes of 80 different animal categories, including 6.8K training images and 1.9K validation images.

**DAWN** [82]) is a vehicle detection dataset that focuses on diverse traffic environment. This dataset contains 1K images from real-traffic environment, including fog, snow, rain and sandstorms. The images are annotated with object bounding boxes of 6 categories.

**MIO-TCD** [5] is an intelligent surveillance dataset for motorized traffic analysis, which contains 137,743 images captured in various times of the day and different periods of the year, and from different viewing angels. This dataset provides bounding box annotations with 11 categories.

**BAAI-VANJEE** [6] is an intelligent surveillance dataset which contains 5K images captured by VANJEE smart base station placed about 4.5m high. The images in this dataset vary in weather and traffic conditions, which are annotated with bounding box annotations of 12 categories.

**DETRAC** [7] is an intelligent surveillance dataset that contains over 14K images captured by a Canon EOS 550D camera at 24 different locations, covering various traffic patterns and conditions including urban highway, traffic crossings and T-junctions. The images in this dataset are annotated with bounding box annotations of 4 categories, including car, bus, van, and others.

**UAVDT** [8]) is a unmanned aerial vehicle detection dataset, which contains about 80K frames from 10 hours videos. The images in this dataset are captured by a unmanned aerial vehicle across various weather conditions (*i.e.*, daylight, night and fog) and multiple camera views (*i.e.*, front-view, side-view and bird-view). This dataset provides bounding box annotations with three categories including car, truck and bus.

# B   Additional Discussion

**Strategy studies for Language Hierarchical Self-training.** Our proposed language hierarchical self-training (LHST) introduces WordNet's language hierarchy [21] to expand the image-level labels and accordingly enables co-regularization between the expanded labels and self-training. We examine the superiority of the proposed LHST by comparing it with "Self-training" [18] and "Direct WordNet Hierarchy Labeling" [21]. "Self-training" is the standard self-training algorithm as in [18] while "Direct WordNet Hierarchy Labeling" denotes directly using the expanded image-level labels (by WordNet) for weakly-supervised detection training. Table 10 reports the experimental results, which show that either "Self-training" [18] or "Direct WordNet Hierarchy Labeling" [21] does not perform well. The reasons are: 1) the box-level pseudo labels in "Self-training" are usually error-prone, making the self-training process unstable and barely improving the performance; 2) the expanded image-level labels in 'Direct WordNet Hierarchy Labeling" are not all reliable, training with which leads to unsatisfying performance. Besides, the combination of 'Self-training" and "Direct WordNet Hierarchy Labeling" still works not very well largely because the direct combination of them does not well address their own drawbacks and limitations. On the other hand, our proposed LHST performs better clearly, as shown in the last row of Table 10. The superior performance of LHST is largely attributed the co-regularization design, which employs self-training to assess and re-weight the expanded labels according to the predicted reliability while enabling the expanded (and re-weighted) labels to regularize self-training by providing richer and flexible supervision (the flexible supervision is achieved by the adaptive re-weighting operation).

Table 10: **Strategy studies for Language Hierarchical Self-training** on zero-shot cross-dataset object detection over object365 dataset.

| Analysis for Language Hierarchical Self- training | AP50 |
| --- | --- |
| Detic [15] | 29.4 |
| Self-training [21] | 29.4 |
| Direct WordNet Hierarchy Labeling [21] | 29.7 |
| Self-training + Direct WordNet Hierarchy Labeling | 29.9 |
| Language Hierarchical Self- training (Ours) | 31.3 |

**Ablation studies of Language Hierarchical Self-training.** As mentioned in the main text, our proposed language hierarchical self-training (LHST) consists of box-leve LHST (i.e., $\mathcal{L}_{box}(F(x))$ in Eq.7 in the main text) and image-level LHST (i.e., $\mathcal{L}_{image}(F(x))$ in Eq.8 in the main text), where image-level LHST could regularize box-level LHST. Here we conduct experiments to investigate this. The experimental results in Table 11 show that Box-level LHST brings clear performance improvements while including Image-level LHST further improves the detection performance, largely becuase Image-level LHST provides stable supervision to regularize Box-level LHST, i.e., Image-level LHST is much more stable as it uses the image-level proposal while the pseudo box labels in Box-level LHST (e.g., the location of pseudo boxes) vary along training iterations and are not very stable.

Table 11: **Ablation studies of Language Hierarchical Self-training.** The experiment setup is zero-shot cross-dataset object detection over Object365 dataset.

| Method | Language Hierarchical Self-training (LHST) | | AP50 |
|---|---|---|---|
| | Box-level LHST | Image-level LHST | |
| Box-Supervised [15] | | | 26.5 |
| Detic [15] | | | 29.4 |
| | ✓ | | 30.5 |
| | ✓ | ✓ | 31.3 |

**Parameter Studies for Language Hierarchical Self- training (LHST).** In generating pseudo box labels in LHST, we filter out a prediction if its max confidence score is lower than the threshold $t$. We study the threshold $t$ by changing it from 0.65 to 0.85 with a step of 0.05. Table 12 reports the experimental results on zero-shot transfer object detection over object365 dataset. We can observe that the detection performance is not sensitive to the threshold $t$.

Table 12: **Parameter Studies for Language Hierarchical Self- training (LHST)** on zero-shot transfer object detection over object365 dataset. We study the thresholding parameter $t$ used in generating pseudo box labels in LHST.

| Threshold $t$ | 0.65 | 0.70 | 0.75 | 0.80 | 0.85 |
|---|---|---|---|---|---|
| AP50 | 31.1 | 31.3 | 31.3 | 31.3 | 31.2 |

**Analysis of discrepancies of the taxonomies of image vs. box categories.** We analyze the mismatch between image-level and box-level categories and how much it could affect the detection performance. As Table 13 shows, the mismatch between image-level and box-level categories varies across datasets, and the proposed DetLH improves more with the increase of mismatch levels.

Table 13: **Analysis of discrepancies of the taxonomies of image vs. box categories.**

| ImageNet-21K | Mismatch Ratio | Baseline (AP50) | DetLH (AP50) | $\Delta$ |
|---|---|---|---|---|
| v.s. Cityscapes | 0.13 | 47.1 | 50.3 | +3.2 |
| v.s. DETRAC | 0.25 | 39.2 | 44.0 | +4.8 |
| v.s. MIO-TCD | 0.27 | 20.6 | 24.5 | +3.9 |
| v.s. African Wildlife | 0.50 | 80.9 | 87.2 | +6.3 |
| v.s. Vistas | 0.67 | 35.6 | 44.0 | +8.4 |
| v.s. Arthropod Detection | 0.86 | 36.7 | 49.0 | +12.3 |

**How effective DetLH deals with noisy labels.** We conduct ablation studies to analyze how effective DetLH deals with noisy labels. Specifically, we compare DetLH with and without using reliability scores (the latter means uniform category weights) over Object365. As Table 14 shows, including the adaptive weighting mechanism (i.e., reliability scores) helps mitigate the label noises effectively.

Table 14: **How effective DetLH deals with noisy labels.**

| Method | AP50 |
|---|---|
| DetLH without reliability score | 31.8 |
| DetLH | 32.5 |

**The impact of using proxy vocabulary.** We conduct experiments on Object365 dataset to compare LHPG with CLIP embeddings only and LHPG with both CLIP embeddings and proxy vocabulary. As Table 15 shows, using

a proxy vocabulary performs clearly better, demonstrating its effectiveness in narrowing the distribution gap between training and test labels.

Table 15: **The impact of using proxy vocabulary.**

| Method | AP50 |
|---|---|
| Baseline | 29.4 |
| LHPG (CLIP embeddings only) | 30.0 |
| LHPG (CLIP embeddings + proxy vocabulary) | 31.0 |

**Comparisons with other semi-supervised WSOD methods.** We conduct experiments on Object365 dataset to compare our DetLH and [88, 89, 90, 91]. As Table 16 shows, DetLH clearly outperforms [a,b,c,d].

Table 16: **Comparisons with other semi-supervised WSOD methods.**

| | Baseline | [88] | [89] | [90] | [91] | DetLH |
|---|---|---|---|---|---|---|
| AP50 | 29.4 | 29.5 | 29.9 | 29.8 | 29.6 | 32.5 |

## C  Additional Comparison

**Comparison with RKD [17].** We note that RKD [17] explores Region-based Knowledge Distillation to better distill knowledge from the CLIP model for weakly-supervised object detection. In this paragraph, we compare our DetLH (i.e., the self-training based method) with RKD (i.e., the knowledge distillation-based method) on zero-shot cross-dataset object detection over Object365 dataset. Table 17 reports the results, which show that our DetLH clearly outperforms RKD [17], indicating the effectiveness the proposed designs in DetLH for zero-shot cross-dataset object detection.

Table 17: Comparison with RKD [17] on zero-shot cross-dataset object detection over Object365 dataset.

| Method | AP |
|---|---|
| RKD [17] | 22.3 |
| DetLH (Ours) | 23.6 |

**Discussion and comparison with visual grounding-based detection methods [92, 93].** We note that GLIP [92] and DetCLIP [93] explore extra visual grounding data to train a open-vocabulary detector. In this paragraph, we compare our DetLH (i.e., the WSOD-based method) with [92, 93] (i.e., the visual grounding-based method) from the perspective of detection efficiency. Table 18 reports the results of run time in millisecond (the run times of GLIP [92] and DetCLIP [93] are acquired from [93].). It shows that our DetLH (i.e., the WSOD-based method) runs much more efficient than the visual grounding-based detection methods (i.e., GLIP [92] and DetCLIP [93]), largely because the visual grounding-based detection methods [92, 93] include a text encoder in their networks. Note we did not compare our DetLH (i.e., the WSOD-based method) with the visual grounding-based method [92, 93] from the perspective of detection accuracy becuase these two types of detection methods use very different training data, e.g., [92] and [93] use visual grounding data. In addition, the WSOD techniques are basically complementary to the visual grounding-based detection techniques [92, 93] because the visual grounding data could be used to further improve the WSOD-based detectors [94]. Similar to [94], we leave this as our future research.

**Other image-level supervision.** We follow Detic [15] to build our proposed DetLH. Therefore, similar to Detic [15], our DetLH can also seamlessly incorporate the free-form caption text as the image-level supervision, i.e., by using the language embeddings of image captions as the detection classifier when training on image-text pair data [15]. In this way, we could further incorporate LAION dataset [95] that includes 400 million image-text pair data into detector training for learning more generalizable object detectors. On the other hand, training over large-scale LAION dataset is computation-intensive and thus we leave this as our future work.

**Comparison with other detection methods [92, 96, 97, 93, 98, 99].** We didn't compare with these methods [92, 96, 97, 93, 98, 99] in the main manuscript because they focus on different topics with different objectives,

Table 18: Efficiency comparison of WSOD-based and visual grounding-based detection methods.

| Method | Types | Run time (ms) |
|---|---|---|
| GLIP [92] w Swin-T | Visual Grounding-based | 8333.3 |
| DetCLIP [93] w Swin-T | Visual Grounding-based | 434.7 |
| DetLH w Swin-B (ours) | WSOD | 46.0 |

training data, backbones and benchmarks. Instead, we follow Detic [15] as both our DetLH and Detic belong to and are claimed as weakly-supervised object detection (WSOD), aiming to using large-scale images and classes (i.e., ImageNet-21K and LVIS) to train a general detector that can work on any detection scenarios. However, [92, 96, 97, 93, 98, 99] are not for WSOD: GLIP and DetCLIP [92, 93] introduce visual grounding and studies how to use grounding data for detection; OWL-ViT [97] focuses on fine-tuning CLIP with standard detection datasets; OmDet and UniDet [98, 99] focus on training with multiple detection datasets. We still managed to benchmark with [92, 96, 97, 93, 98, 99], i.e., GLIP [92], GLIPv2 [96], OWL-ViT [97], DetCLIP [93], OmDet [98] and UniDet [99]. As our method and [92, 96, 97, 93, 98, 99] use various different datasets in evaluations, the benchmarking below is on the shared one, i.e., Pascal VOC (in AP).

| GLIP-L | GLIPv2-B | OWL-ViT | DetCLIP | OmDet | UniDet | Ours |
|---|---|---|---|---|---|---|
| 61.7 | 62.8 | 60.3 | 56.7 | 60.8 | 60.1 | 64.4 |

Note Florence [100] is not included as it is a foundation model that uses a very large backbone (CoSwin-H) and very large customized training data (FLD-900M and FLOD-9M). Besides, we compared our DetLH with GLIP [92] and DetCLIP [93] as in Table 18, which shows DetLH (i.e., the WSOD-based method) runs much more efficient (about 10 times) than the visual grounding-based detection methods (i.e., GLIP and DetCLIP).

**Comparison on ODinW [92].** We note that ODinW [92] also benchmarks cross-dataset generalization. We benchmark on ODinW and the results below (averaged on 35 datasets in ODinW) show that our DetLH works effectively on ODinW. Note GLIP obtains higher accuracy because it introduces visual grounding and involves Language Encoder in inference. Without those extra modules, our DetLH runs much faster (over 10 times) than GLIP as discussed in Table 18.

| WSDDN | YOLO9000 | DLWL | Detic | GLIP | OmDet | Ours |
|---|---|---|---|---|---|---|
| 14.9 | 16.3 | 17.1 | 17.3 | 19.7 | 16.0 | 18.2 |

**Open-vocabulary benchmark.** We did not benchmark on open-vocabulary LVIS/COCO (both divide a single-dataset vocabulary into base and novel classes to mimic and benchmark vocabulary generalization), because our work aims to leverage large-scale images and classes (i.e., 21K classes) to train a general detector that can work on any detection scenarios. Cross-dataset generalization benchmark fits this objective better and is more general and challenging than open-vocabulary benchmark that tackles base and novel classes within a single dataset.

**Comparison with class hierarchy methods [101, 70] on OpenImages [102].** We benchmark with other methods that use class hierarchy, including [101] and [70]. As shown below, our DetLH performs clearly better than hierarchy-aware losses in [101, 70] due to our designed co-regularization as detailed in the manuscript. Note we use OpenImages V7 (Oct 2022) instead of 2019 version.

| WSDDN | YOLO9000 | DLWL | Detic | [101] | [70] | Ours |
|---|---|---|---|---|---|---|
| 14.8 | 15.8 | 16.2 | 16.4 | 16.5 | 16.5 | 17.6 |

**Results of WSDDN, YOLO9000 and DLWL.** Note Detic implemented WSDDN, YOLO9000 and DLWL and we directly adopted Detic's implementation in evaluations. The gains in our reported results are different as we evaluated on more challenging datasets and benchmark: 1) The results in our Table 7 in the main manuscript are averaged over 14 datasets while those in Table 1 of Detic are on a single dataset LVIS; 2) Our Table 7 in the main manuscript is cross-dataset generalization benchmark while Table 1 in Detic is open-vocabulary LVIS.

# D  Additional Qualitative Result and Comparison

We provide qualitative results of zero-shot cross-dataset object detection for various detection tasks. As shown in Figures 3- 7. our DetLH produces good detection results consistently across different detection tasks, showing DetLH still works effectively under large cross-dataset gaps in data distribution and vocabulary.

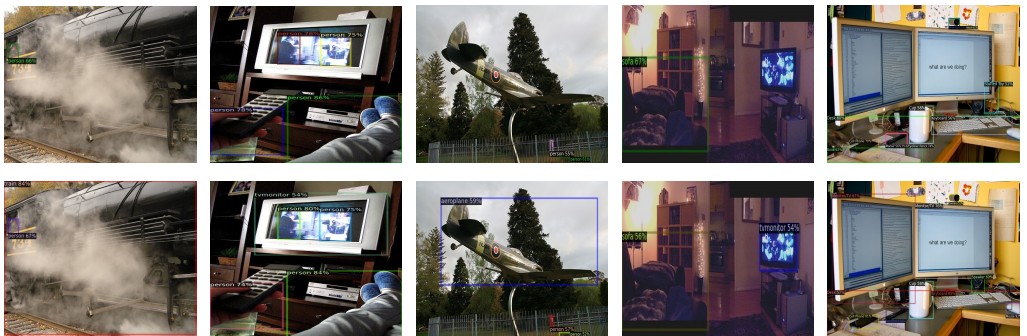

Figure 3: Qualitative results of DetLH over zero-shot cross-dataset object detection for common objects. Zoom in for details. Top: Detic [15]. Bottom: DetLH (Ours).

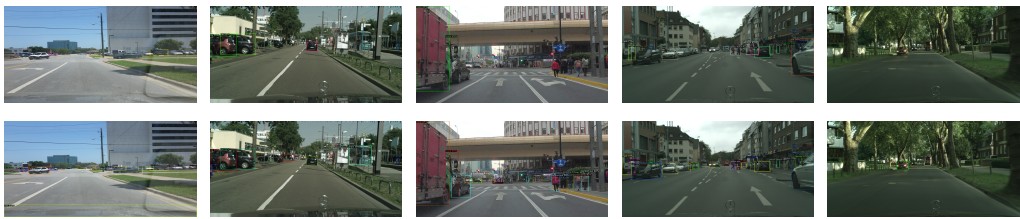

Figure 4: Qualitative results of DetLH over zero-shot cross-dataset object detection for autonomous driving. Zoom in for details. Top: Detic [15]. Bottom: DetLH (Ours).

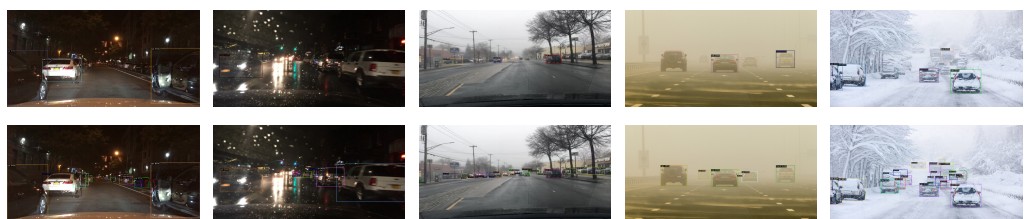

Figure 5: Qualitative results of DetLH over zero-shot cross-dataset object detection under different weather and time-of-day conditions. Zoom in for details. Top: Detic [15]. Bottom: DetLH (Ours).

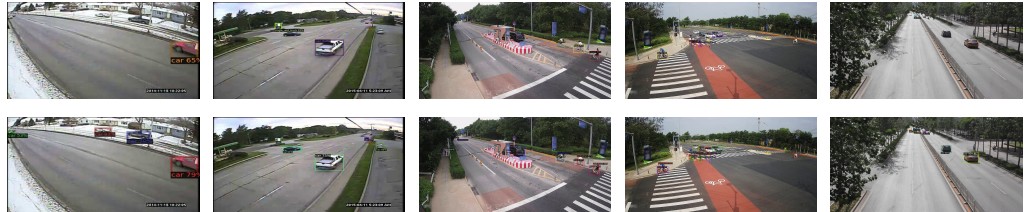

Figure 6: Qualitative results of DetLH over zero-shot cross-dataset object detection for intelligent surveillance. Zoom in for details. Top: Detic [15]. Bottom: DetLH (Ours).

# E   Broader Impacts and Limitations

**Broader Impacts.** This work strives for exploiting weakly-supervised object detection (WSOD) to learn generalizable detectors by addressing the image-level label mismatch issue. We propose to incorporate image-level supervision with self-training for learning generalizable detectors, aiming to benefit from self-training while effectively making use of image-level weak supervision. Our proposed technique provides great advantages by avoiding the need of massive object-level annotations and allowing learning effective and generalizable detectors with image-level supervision. It thus makes a very valuable contribution to the computer vision

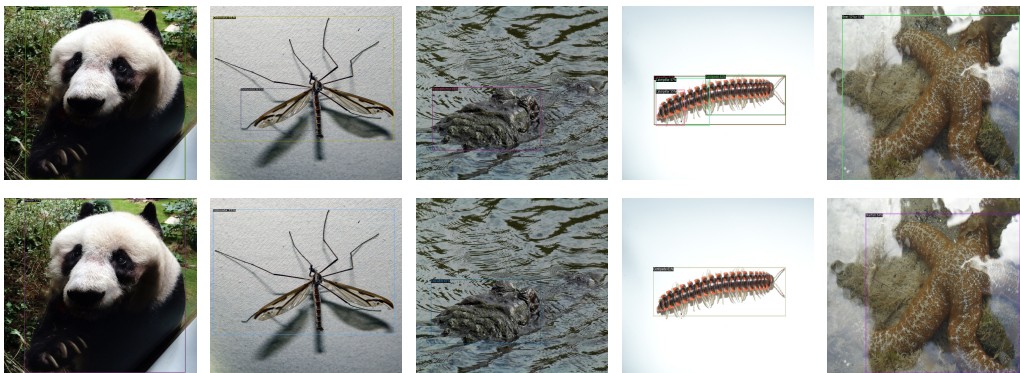

Figure 7: Qualitative results of DetLH over zero-shot cross-dataset object detection for Wildlife Detection. Zoom in for details. Top: Detic [15]. Bottom: DetLH (Ours).

research community by providing a novel and efficient weakly-supervised object detection method. The feature of scaling detectors with image-level labels enables effective and generalizable detectors that could work well in various downstream tasks, broadening the applicability of object detectors significantly.

**Limitations.** As discussed in Sections 3 and 4 of the main text and Section Dataset and Implementation Details in the appendix, our proposed WSOD method adopts ImageNet-21K with image-level labels to scale up detectors. It avoids the need of massive object-level annotations and allowing learning effective and generalizable detectors with image-level supervision. At the other end, we could further scale up detector training by involving the recent image-text pair data for WSOD training, which may further improve the performance significantly. We will investigate how to involve the recent image-text pair data for WSOD training in our future work.

