# OpenReview forum: "Open-Vocabulary Object Detection via Language Hierarchy"
_NeurIPS.cc/2024/Conference — NeurIPS 2024 poster_

### Official Review · Reviewer_VMio · 2024-07-11

**Soundness:** 4
**Presentation:** 4
**Contribution:** 3
**Rating:** 7
**Confidence:** 3

**Summary:**

The paper proposes a novel method for leveraging image-level annotations for object detection pretraining, specifically for zero-shot and open vocabulary detection settings. The proposed method, DetLH, leverages self-training to generate pseudo-labelled object proposals, and then adjust the pseudo-label for each box according to the image's class and the classes in WordNet that are hierarchically related to it. Furthermore, the authors propose a method for prompt generation that does not require additional training. The proposed method is evaluated extensively and achieves state-of-the-art results in relevant zero-shot and open-vocabulary object detection benchmarks.

**Strengths:**

1. The paper is well written and its key ideas clearly communicated.
2. The proposed method is well presented and novel, and constitutes an improved way to leverage labelled object centric images to facilitate object detector training.
3. The conducted experiments are extensive and demonstrate that the proposed method consistently achieves state-of-the-art performance in various settings and datasets.

**Weaknesses:**

1. Unless I am mistaken, there is no ablation study related to the impact of the reliability score. I would be very interested to see how it affects performance.
2. While justifications are provided in the appendix, I believe more comparisons should have been included with relative methods (L661-671). More importantly, I believe that the ommision of evaluations with LVIS/COCO is a significant issue, and I am not convinced by the authors stated reason (L683-687) that they focused on better suited datasets. The paper has extensive experiments with many datasets, and it is confusing to me why they chose to leave out what is arguably the most common dataset for detection tasks.

**Questions:**

My primary concern regarding the paper is related to the evaluations: a) Whereas the paper is a clear improvement over Detic, it avoids extensive comparisons with more recent methods (Detic being a ECCV 2022 paper) such as the ones included in the paper (L661-671), and b) evaluations with LVIS, a shared benchmark for most relative papers, are not conducted.

Overall, however, I am very satisfied with the paper's contribution and I believe that the conducted experiments adequately demonstrate the proposed method's effectiveness. My main reason for not giving a higher rating is because LVIS/COCO results are not included, which would be informative and are significant to facilitate comparisons with existing and future relevant works.

**Limitations:**

The authors have adequately addressed the relevant limitations.

---

> ### Author Rebuttal · Authors · 2024-08-06
>
> **[Response 1] The affect of the reliability score:**
>
> Thank you for your suggestion. As suggested, we conduct the new ablation study to examine the effect of the reliability score in the proposed DetLH. As the table below shows (on Object365), DetLH effectively mitigate the label noises with the design of reliability score.
>
> | | AP50|
> |:----:|:----:|
> |DetLH without reliability score | 31.8|
> |DetLH | 32.5 |
>
> **[Response 2] More comparisons:**
>
> We would clarify that we did not provide much comparisons with those methods as mentioned in Lines 661-671 because they focus on different tasks with different objectives (also use different training data, backbones and benchmarks). We largely focus on benchmarking DetLH with Detic as both methods work on the same task WSOD and use the same training data (ImageNet-21K and LVIS) and backbone. As suggested, we will provide more comparisons between DetLH and those detectors and include the comparisons in the updated manuscript or appendix.
>
> In addition, we would clarify that we focus on cross-dataset zero-shot evaluations [48, 68] (i.e., evaluation on unseen datasets like CLIP) as discussed in Lines 117-119. We did not benchmark on LVIS/COCO (both use the same data but different annotations) since we used LVIS/COCO in network training.
>
> As suggested, we benchmark DetLH on LVIS and COCO as shown below and will include these results in the updated manuscript or appendix.
>
> |LVIS Dataset | AP50|
> |:----:|:----:|
> |MosaicOS  |28.3 |
> |CenterNet |34.9 |
> |AsyncSLL |36.0 |
> |SeesawLoss  |37.3  |
> |Baseline |40.7 |
> |Detic |41.7 |
> |**DetLH** | **42.2** |
>
>
> |COCO Dataset | AP50|
> |:----:|:----:|
> |Baseline |39.3 |
> |Self-training |39.5 |
> |WSDDN |39.9 |
> |DLWL |42.9 |
> |Detic |44.7 |
> |**DetLH** |**45.3** |

---

> > ### Comment · Reviewer_VMio · 2024-08-11
> > **Reply to authors**
> >
> > I thank the authors for addressing the issues raised by my review.
> >
> > I would like to also see what other reviewers think about the responses to the issues they raised, but I am satisfied by the authors’ response to my review and am inclined to recommend acceptance (i.e. raise my score to 7 after the discussion).

---

> > > ### Author Response · Authors · 2024-08-13
> > >
> > > Dear Reviewer VMio,
> > >
> > > Thank you for your encouraging feedback and positive evaluation for our work. We will include the new ablation studies and the more comparisons in the main paper in the revised version. We sincerely appreciate your constructive comments, which have strengthened our paper.
> > >
> > > Best regards,
> > >
> > > Authors

---

### Official Review · Reviewer_PjbE · 2024-07-13

**Soundness:** 3
**Presentation:** 3
**Contribution:** 2
**Rating:** 6
**Confidence:** 4

**Summary:**

This paper focuses on scaling the detectors’ vocabulary with image-level weak supervision. To better leverage the image-level labels in this task, this paper introduces a language hierarchical self-training (LHST) framework that incorporates language hierarchy (i.e., WordNet) with self-training. In addition, the authors introduce a language hierarchical prompt generation (LHPG) method enabling open-vocabulary object detection. The proposed method outperforms baselines across 14 commonly used benchmarks. Comprehensive ablation experiments validate the effectiveness and generalization of the proposed method.

**Strengths:**

+ This paper is well organized and written. The overall paper is easy to follow and technically sound.
+ On extensive benchmarks, the proposed method achieves non-trivial performance improvement and shows strong generalization.

**Weaknesses:**

- The necessity of the investigated problem (i.e., scaling the vocabulary of the detectors with image labels) should be further stated, as the recent MLLM-based detection/grounding methods inheritly have strong zero-shot capabilities.
- The proposed method is somewhat trivial and heuristic, where only the usage of WordNet is new and interesting. Most techniques are from existing WSOD and semi-supervised object detection methods. It would be better to highlight the novelty.
- Several semi-supervised WSOD methods trained with both instance- and image-level annotations are missing. It would be better to review them for completeness.
[a] UniT: Unified Knowledge Transfer for Any-Shot Object Detection and Segmentation, CVPR’21
[b] Cyclic Self-Training with Proposal Weight Modulation for Cross-Supervised Object Detection, TIP’23
[c] H2FA R-CNN: Holistic and Hierarchical Feature Alignment for Cross-Domain Weakly Supervised Object Detection, CVPR’22
[d] DOCK: Detecting Objects by transferring Common-sense Knowledge, ECCV’18
- In Line 149, the authors claimed ‘a single-label annotation could be expanded into a multi-label annotation’. It would be better to give some visualization examples. In Line 164, authors claimed ‘the labels expanded by WordNet are not all reliable’. Giving some examples would be helpful to improve the readability.
- Why only the AP results of oracles are reported on Object 365?
- How about the efficiency of WordNet in training and inference?

**Questions:**

Please refer weakness.

**Limitations:**

The authors have discussed the limitations, and this works does not show potential negative societal impact.

---

> ### Author Rebuttal · Authors · 2024-08-06
>
> **[Response 1] The necessity of the investigated problem:**
>
> Thank you for your suggestion. We would clarify that, compared with recent MLLM-based detection/grounding methods, our approach of scaling the vocabulary of detectors using image labels offers significant advantages in training efficiency. As shown in Table 14 (as copied and posted below), DetLH is up to ten times more efficient than current visual grounding/MLLM-based methods such as GLIP and DetCLIP. The superior efficiency is critical to the scalability of detectors in many practical tasks.
>
>
> | Method | Types | Run time (ms)|
> |:----:|:----:|:----:|
> GLIP w Swin-T |Visual Grounding-based |8333.3|
> DetCLIP w Swin-T |Visual Grounding-based |434.7 |
> DetLH w Swin-B (ours) | WSOD |46.0|
>
> **[Response 2] The novelty of the proposed method:**
>
> Thank you for your suggestion! We would share that, different from existing WSOD and semi-supervised WSOD methods, DetLH tackles the image-to-box label mismatch by introducing language hierarchy into self-training: 1) it introduces WordNet's language hierarchy to expand the image labels and accordingly enables co-regularization between the expanded labels and self-training. As shown in Table 10 (in the submitted manuscript), just introducing language hierarchy or simply combing self-training and language hierarchy does not work without our co-regularization design; 2) it introduces language hierarchy into prompt generation to bridge the vocabulary gaps between detector training and testing, leading to better prompt generation and detection performance. Extensive experiments validate the effectiveness of our designs in DetLH. We will highlight the above points in the updated manuscript.
>
> **[Response 3] Comparisons with other semi-supervised WSOD methods:**
>
> Thanks for sharing [a,b,c,d]. We will review them and highlight how DetLH differs in the revised paper. As the table below shows, DetLH clearly outperforms [a,b,c,d] on Object365.
>
> | | Baseline | [a] | [b] | [c] | [d] | DetLH|
> |:----:|:----:|:----:|:----:|:----:|:----:|:----:|
> |AP50 |29.4 |29.5 |29.9|29.8|29.6|**32.5**|
>
>
> **[Response 4] Visualizations of label expanding:**
>
> Thank you for your suggestion! **Figure 1** in the attached PDF shows that the labels expanded by WordNet are not all reliable. We will include the suggested visualization in the updated manuscript or appendix.
>
> **[Response 5] Missing oracle values on Object365 in Table 1:**
>
> We reported AP value only for Object365 as we referenced to the Detic paper which only reports AP value. For Pascal VOC, we trained the model to get the oracles of all AP metrics. We will train models on Object365 to obtain the complete AP values and update them in the updated manuscript.
>
> **[Response 6] The efficiency of WordNet in training and inference:**
>
> |  | Training Speed (second/image) | Inference Speed (second/image)|
> |:----:|:----:|:----:|
> |Detic (w/o WordNet language hierarchy)| 0.7009 | 0.045|
> | Our DetLH |0.7110| 0.046|

---

> > ### Comment · Reviewer_PjbE · 2024-08-13
> >
> > Thanks for the efforts in rebuttal. Your response has addressed my concerns.
> > After reading other reviewers' comments and the response, I would like to raise my rating.

---

> ### Author Response · Authors · 2024-08-13
>
> Dear Reviewer PjbE,
>
> Thank you for your insightful feedback. We have carefully considered your questions and suggestions and have addressed them accordingly. We sincerely appreciate your constructive comments, which have helped strengthen our paper. As the discussion phase is nearing its conclusion, we would appreciate it if you could let us know if there are any additional questions or suggestions.
>
> Best regards,
>
> Authors

---

### Official Review · Reviewer_TMzm · 2024-07-30

**Soundness:** 2
**Presentation:** 3
**Contribution:** 2
**Rating:** 4
**Confidence:** 3

**Summary:**

* The authors propose to expand the training data for object detection using classification datasets through two main contributions:

1. Combine a language hierarchy with self-training to extend the training dataset, while minimizing label noise by re-weighting categories with reliability scores.
2. Bridge the vocabulary gaps between training and testing using a vocabulary proxy (WordNet) improving zero-shot learning.

* Extensive experiments in multiple datasets are provided showing an improvement over the state-of-the-art.

**Strengths:**

1.The authors clearly exposed their ideas and the paper is reasonably well written.

2.Even though using a language hierachy to improve object recognition is an old idea, [Marszalek, et al. Semantic Hierarchies for Visual Object Recognition] it's appealing to see it applied to a modern, large-scale, object detection setting. Moreover, the combination with self-training is original and well justified.

3.The results show a significant improvement over the state-of-the-art.

4.The evaluation is very extensive and performed over an impressive quantity of diverse datasets.

**Weaknesses:**

1. The experimental section and ablation do not clearly show if the authors' assumptions are correct.
* The idea of the paper heavily relies on the assumption that there are discrepancies on the taxonomy levels of the categories of image and detection datasets. This assumption may hold for some datasets but not for others and it should have a significant impact on the method's results. I miss a section presenting an analysis of the taxonomies of image vs box categories, and how it may impact peformance.

2. Discussion on the specific impact of each contribution to the final result is lacking
* The ablation study (Section 4.2) does not seem to add much value to the experimental discussion. First, showing the performance increase over the baseline [68] seems misleading when the authors compare DetLH against only-box supervision, since the work of [68] already proposes to expand detector (boxed) training data with image-classification data. The work of [68] already demonstrated that adding image-based training data improves detection results, and it should not be the goal of the ablation section to show this. A far more interesting baseline could have been to train DetLH with uniform category weights to demonstrate how DetLH deals with noisy labels.
* Another important ablation experiment would be to analize the impact of narrowing the gap between training and testing label distributions. The CLIP embeddings would already help in this regard, but how impactful is the use of a proxy vocabulary on top of it is not clearly quantified.

3. Wording is sometimes ambiguous or could lead to contribution misunderstandings:
* For instance the authors say "These detectors usually yield constrained detection peformance when trained only with detection datasets... [68]", when the work of [68] is actually trained with both detection and classification data. If the authors want to point out that [68] already suggested this, it should be more explicitly stated. Even more so, if it's followed with a claim that the current method overcomes such limitations: "DetLH introduces large-scale image-level datasets to enlarge the data [...] in detector training.", which is misleading as this is not something the present work introduces, but the work of [68].

**Questions:**

* Futher analysis and validation of main assumption. How do the category taxonomy levels differ across datasets?
* Where does the performance gain come from? Robustness to label noise or vocabulary proxy?
* In general in the experimental section I miss more insight and discussion on results, experimental setup and data setup. I feel that most discussion repeatedly states the obvious (performance increases due to leveraging larger amouts of data). Whenever more specific insight is provided, it is very speculative. For example, authors state that performance gains in Wildlife are due to the "introduction of the language hierarchy", but there are no specific experiments designed to proof such claim.

**Limitations:**

Yes, the limitations and potential negative impact are addressed.

---

> ### Author Rebuttal · Authors · 2024-08-06
>
> **[Response 1] Analysis of discrepancies of the taxonomies of image vs. box categories:**
>
> Thank you for pointing out this issue! As suggested, we analysed the mismatch between image-level and box-level categories and how much it could affect the detection performance. As the table below shows, the mismatch between image-level and box-level categories varies across datasets, and the proposed DetLH improves more with the increase of mismatch levels. We will include a subsection to discuss this issue in the updated paper.
>
> |ImageNet-21K |Mismatch Ratio| Baseline (AP50) | DetLH (AP50) |  $\Delta$ |
> |:----:|:----:|:----:|:----:|:---:|
> v.s. Cityscapes	|0.13 	|47.1	|50.3	|+3.2|
> v.s. DETRAC	|0.25 	|39.2	|44.0	|+4.8|
> v.s. MIO-TCD	|0.27 	|20.6	|24.5	|+3.9|
> v.s. African Wildlife	|0.50 	|80.9	|87.2	|+6.3|
> v.s. Vistas	|0.67 	|35.6	|44.0	|+8.4|
> v.s. Arthropod Detection|	0.86 	|36.7	|49.0	|+12.3|
>
> **[Response 2] More ablation studies:**
>
> Thank you for your comment! As suggested, we conduct new ablation studies to analyse how effective DetLH deals with noisy labels. Specifically, we compare DetLH with and without using reliability scores (the latter means uniform category weights) over Object365. As the table below shows, including the adaptive weighting mechanism (i.e., reliability scores) helps mitigate the label noises effectively.
>
> | | AP50|
> |:----:|:----:|
> DetLH without reliability score | 31.8|
> DetLH | 32.5 |
>
> Regarding the impact of using proxy vocabulary, we conduct new experiments on Object365 dataset to compare LHPG with CLIP embeddings only and LHPG with both CLIP embeddings and proxy vocabulary. As the table below shows, using a proxy vocabulary performs clearly better, demonstrating its effectiveness in narrowing the distribution gap between training and test labels.
>
> | | AP50|
> |:----:|:----:|
> Baseline | 29.4|
> LHPG (CLIP embeddings only) | 30.0 |
> LHPG (CLIP embeddings + proxy vocabulary) | 31.0 |
>
> **[Response 3] Misleading Claims:**
>
> Thank you for your suggestion! To avoid confusion, we will rephrase the text "These detectors usually yield constrained detection... [68]" to "As reported in [68], detectors trained solely on detection datasets often exhibits constrained performance.", and the claim "DetLH introduces large-scale image-level datasets  [...]." to "Similar to [68], DetLH follows semi-supervised WSOD and introduces large-scale image-level datasets for detector training.".
>
> **[Response 4] More insights and analysis should be provided:**
>
> Thank you for pointing out this issue. We would clarify that we provided extensive discussion and related insights in the appendix (e.g., Section B Additional Discussion, and Section C Additional Comparison), including more ablation studies, strategy studies, parameter studies, comparisons, etc. In addition, we benchmark DetLH across multiple datasets and discuss its generalization ability and parameter sensitivity In Section 4.1 Comparison with the SOTA and Section 4.3 Discussion. We will revise the two Sections to remove repetitive text and include some insight analysis from the appendix.
>
> Regarding Wildlife datasets, our statement is largely based on the intuition that these datasets with many fine-grained categories should suffer from clear vocabulary mismatch, and the language hierarchy introduced by DetLH should help mitigate such mismatch and improve the detection. The new experiments in Response 1 show that wildlife datasets do suffer from more severe mismatch and DetLH helps mitigate the mismatch effectively.

---

### Author Rebuttal · Authors · 2024-08-06

We thank the reviewers for your positive comments on our work. In particular, it is encouraging that the reviewers acknowledge that 1) our work is novel and appealing [TMzm,VMio]; 2) the proposed method is effective [TMzm,PjbE,VMio]; 3) the evaluation is extensive [TMzm,PjbE,VMio] and 4) the paper is well written [TMzm,PjbE,VMio].

We address the questions and concerns raised by each reviewer point-by-point in the respective threads. Additionally, attached please find a PDF file that contains the figure that is related to the rebuttal.

---

### Comment · Reviewer_TMzm · 2024-08-12
**Reply to authors**

Thanks to the authors for the thorough clarification of my cocerns. I'm happy to boost my score to "Weak Accept" as long as the updated ablation study, as well as further discussions regarding experiments (and taxonomy analysis) are included in the main paper. In my opinion those are important points that should not be left in the supplementary material.

---

> ### Author Response · Authors · 2024-08-13
>
> Dear Reviewer TMzm,
>
> Thank you for your encouraging feedback and positive evaluation for our work. We will include the updated ablation study, as well as further discussions regarding experiments (and taxonomy analysis) in the main paper in the revised version, and we agree that these are important points that should not be left in the supplementary material. We sincerely appreciate your constructive comments, which have strengthened our paper.
>
> Best regards,
>
> Authors

---

### Decision · Program_Chairs · 2024-09-25

**Decision:**

Accept (poster)

**Comment:**

This submission proposes to expand the training data for object detection using image-level weak supervision from classification datasets, and leverages a word ontology (wordnet).

Reviewers agreed that:
1. The paper was generally well-written and easy to understand
2. The proposed method yields meaningful improvements over SOTA.

There were initial concerns about:
1. The thoroughness of ablations and experimental analysis (including suggestions to move some experiments to the main section)
2. The proposed method being somewhat trivial.

Both concerns were addressed during the discussion period and all 3 reviews ultimately recommend weak accept or accept. The AC sees no reason to override this unanimous recommendation. The authors are asked to incorporate the requests from reviewers into the final version of the paper.